# Can Hematological Inflammatory Indices Be Used to Differentiate Modic Type 1 Changes from Brucella Spondylodiscitis?

**DOI:** 10.3390/medicina60071131

**Published:** 2024-07-14

**Authors:** Volkan Şah, Ali İrfan Baran

**Affiliations:** 1Department of Sports Medicine, Medical Faculty, Van Yüzüncü Yıl University, Van 65090, Turkey; 2Department of Infectious Diseases and Clinical Microbiology, Medical Faculty, Van Yüzüncü Yıl University, Van 65090, Turkey; a.irfanbaran@gmail.com

**Keywords:** brucellosis, spondylodiscitis, blood cell count, inflammation, biomarkers

## Abstract

*Background and Objectives*: Differentiation between brucella spondylodiscitis and Modic type I changes (MC1) includes difficulties. Hematological inflammatory indices (HII) such as neutrophil to lymphocyte ratio (NLR) and aggregate index of systemic inflammation (AISI) are suggested as indicators of inflammation and infection and have diagnostic, prognostic, and predictive roles in various diseases. This study aimed to evaluate differences between brucella spondylodiscitis and MC1 in terms of HII. *Materials and Methods*: Thirty-five patients with brucella spondylodiscitis and thirty-seven with MC1 were enrolled in the study. Brucella spondylodiscitis and MC1 were diagnosed by microbiological, serological, and radiological diagnostic tools. HII (NLR, MLR, PLR, NLPR, SII, SIRI, AISI) were derived from baseline complete blood count. *Results*: The two groups were similar for age (*p* = 0.579) and gender (*p* = 0.092), leukocyte (*p* = 0.127), neutrophil (*p* = 0.366), lymphocyte (*p* = 0.090), and monocyte (*p* = 0.756) scores. The Brucella spondylodiscitis group had significantly lower pain duration (*p* < 0.001), higher CRP and ESR levels (*p* < 0.001), and lower platelet count (*p* = 0.047) than the MC1 group. The two groups had similarity in terms of HII: NLR (*p* = 0.553), MLR (*p* = 0.294), PLR (*p* = 0.772), NLPR (*p* = 0.115), SII (*p* = 0.798), SIRI (*p* = 0.447), and AISI (*p* = 0.248). *Conclusions*: Increased HII can be used to differentiate infectious and non-infectious conditions, but this may be invalid in brucellosis. However, pain duration, CRP and ESR levels, and platelet count may be useful to distinguish brucella spondylodiscitis from MC1.

## 1. Introduction

Modic changes are different signal intensities with different histopathological contents in the disk–vertebral region seen on magnetic resonance imaging (MRI). The signal intensities for Modic type 1 changes (MC1s) are low and high on the T1 and T2 weighted MRI sequences, respectively; for MC2s, they are high on the both T1 and T2 weighted MRI sequences; and for MC3s, they are low on the both T1 and T2 weighted MRI sequences (Table 1) [1,2]. Histopathologically, MC1s represent bone marrow edema and inflammation; MC2s represent the fatty degeneration of the bone marrow; and MC3s represent subchondral bone sclerosis [2]. In addition, histomorphometric examinations of the bone structure and activity have demonstrated that MC1s have high bone turnover; MC2s have low bone formation; and MC3s have stable sclerosis (Table 1) [3]. MC3s are more rare, but the types of Modic changes can transform into each other over time, or they can be seen as combined [1]. The association of Modic changes with low back pain has been shown [4], and MC1s are more painful than other types [5]. The fact that MC1s are more painful has been attributed to their inflammatory properties [2,6]. It has been suggested that MC1s are related to inflammatory conditions [1], traumatic harm [7,8], past infection destruction [9,10], and autoimmune reaction [11], while MC2s are associated with mechanical load and systemic actions [1,7,8]. However, there is no specific therapeutic approach for Modic changes, and it is accepted that non-surgical and personalized treatments according to their types should be considered [1,8].

Brucella spondylodiscitis is the predominant and most severe form of musculoskeletal brucellosis [12,13]. Its clinical manifestations include back pain, myalgia, arthralgia, fatigue, and fever [14]. However, these non-specific symptoms are causes of misdiagnosis or belated diagnosis in patients with Brucella spondylodiscitis [14,15]. It has been recommended that spinal brucellosis should be noted in patients with chronic back pain, especially in endemic regions [16,17], and also in patients with fever of unknown origin, especially in non-endemic regions [18]. Similar to clinical features, laboratory results are also non-specific; for instance, high levels of ESR, CRP, as well as increased or decreased levels of leukocytes, lymphocytosis, and thrombocytopenia may be seen [14,19]. Similar to clinical features and laboratory results, MRI findings are also non-specific and follow the classic infection/inflammation appearance, including low signal intensity on the T1 and high signal intensity on the T2 weighted images [14]. This refers to bone marrow edema and is parallel to the signals of MC1 [9,14,20]. The most specific and definitive diagnosis of Brucella spondylodiscitis is based on microbiological analyses such as positive culture, ≥4-fold increase in antibody titer after at least two weeks, ≥1:160 antibody titer in agglutination tests, PCR-based DNA determination, and ELISA-based specific antibody detection [14,21]. The treatment of Brucella spondylodiscitis includes a combination of a long course of two or three antibiotics [14,22], but cases with neurological signs and treatment-resistant patients require surgical intervention [14].

Brucella spondylodiscitis and MC1 are similar to each other in terms of some features. Both MC1 and Brucella spondylodiscitis mostly involve the lumbar spine [15,23,24]. Both of these conditions manifest as back pain [5,14]. In addition, both of the two disorders display similar MRI findings (low and high on the T1 and T2 weighted images, respectively), which are infectious/inflammation/bone marrow edema reflections [1,14,25]. These similarities can lead to some challenges in differentiation between brucella spondylodiscitis and MC1 [25]. Therefore, a detailed MRI examination is required to differentiate these two entities. For example, a solid/irregular endplate contour on T1-weighted images is a strong indicator of MC1, while its absence is highly significant for infectious spondylodiscitis [26]. On the other hand, an almost intact vertebral architecture is found in brucella spondylodiscitis [27]. In addition to these traditional MRI findings, more advanced examinations that can differentiate these two entities may be required. For example, the “claw sign” finding on diffusion-weighted MRI has been proposed to distinguish MC1 from infectious spondylodiscitis in the lumbar spine [25].

Although specific MRI and microbiological examinations are useful tools in diagnosis and differential diagnosis, they are advanced, time-consuming, costly, impractical, and hardly accessible methods. Therefore, there is a need to evaluate more appropriate methods in differentiation between brucella spondylodiscitis and MC1. With this understanding and perspective, we aimed to address the value of some hematological indices [28,29,30] to differentiate MC1 from Brucella spondylodiscitis.

Hematological inflammatory indices (HII) (Table 2) are currently very popular and have diagnostic [31,32], prognostic [29,33], and predictive [30,34] roles in various diseases. Considering their promising roles, we hypothesized that HII may have a distinctive value between brucella spondylodiscitis and MC1. If the hypothesis is valid, early diagnosis-differential diagnosis–treatment processes may become easier and more successful. Given that HIIs are faster, practical, simpler, inexpensive, and easily accessible indicators, they may be more appropriate tools in differentiation between brucella spondylodiscitis and MC1. Therefore, the present study is focused on detecting these differences, especially in the simple clinical and inflammatory parameters between brucella spondylodiscitis and MC1.

## 2. Materials and Methods

This is a retrospective comparative study focusing on distinguishing between brucella spondylodiscitis and MC1 considering HII. Patients’ data were obtained from Hospital Information Systems between 2020 and 2024. The Hospital’s Ethics Committee approved the study protocol (reference number 2024/3-45, date of approval: 29 February 2024). In accordance with the principles of the Declaration of Helsinki, the study was designed, carried out, and completed. Also, the study was registered on ClinicalTrials.gov with the number NCT06432530.

A total of 35 patients with brucella spondylodiscitis and 37 MC1 were enrolled in the study. Diagnoses of brucella spondylodiscitis and MC1 were supported by microbiological, serological, and radiological diagnostic tools. A positive culture from any specimen was considered a definitive diagnosis of brucellosis. Patients’ hematological parameters were recorded, and HII (NLR, MLR, PLR, NLPR, SII, SIRI, AISI) were derived from baseline complete blood count (CBC).

### 2.1. The Inclusion Criteria

Based on the diagnostic tools and criteria [1,2,14,21], cases diagnosed with lumbar brucella spondylodiscitis or lumbar MC1 in the past 5 years and who had simultaneous lumbar MRI, CBC test, C-reactive protein (CRP), and erythrocyte sedimentation rate (ESR) results, and aged 18–65 years were selected to yield a study population.

### 2.2. The Exclusion Criteria

On the other hand, cases with inadequate data, aged <18 or >64 years, other infectious spondylodiscitis types than brucella, other types of Modic changes than MC1, and other non-infectious conditions such as rheumatic spondylodiscitis (ankylosing spondylitis or Andersson lesion) were excluded from the study. Also, previous or recurrent brucella spondylodiscitis, involving spinal levels other than the lumbar spine, were exclusion causes.

The two groups were statistically assessed and compared for baseline features such as age, gender, symptom duration, CRP, ESR, CBC, and HII derived from the CBC cells.

Statistical analyses were performed using IBM SPSS version 20.0 (IBM Corp., Armonk, NY, USA). The Kolmogorov–Smirnov test was used to assess the normality states of continuous variables. The SIRI and AISI scores in the brucella spondylodiscitis group had a non-normal distribution. Accordingly, for these variables, the Mann–Whitney U test was applied in the comparisons between the two groups. For normally distributed variables, the independent *t*-test was applied in the comparative statistics. Fisher’s exact test and the Pearson’s chi-squared test were used for the categorical variables. A *p*-value of ≤0.05 was accepted as a statistically significant level.

## 3. Results

### 3.1. Comparison of the Groups in Age, Gender, Pain Duration, CRP, ESR, and Spinal Levels

We found that the groups were similar in terms of age (*p* = 0.579) and gender (*p* = 0.092). However, the brucella spondylodiscitis group had significantly higher scores of lumbar pain duration (*p* < 0.001), CRP (*p* < 0.001), and ESR (*p* < 0.001) levels than the MC1 group. There was also a significant difference between the two groups in the involved spinal levels (*p* = 0.030) (Table 3).

### 3.2. Comparison of the Groups in Complete Blood Cells

There were no statistically significant differences between the groups in terms of leukocyte (*p* = 0.127), neutrophil (*p* = 0.366), lymphocyte (*p* = 0.090), and monocyte (*p* = 0.756) scores. However, the brucella spondylodiscitis group had a significantly lower platelet count than the MC1 group (*p* = 0.047) (Table 4).

### 3.3. Comparison of the Groups in HII

We found that the groups had similarities in terms of investigated HII, including NLR (*p* = 0.553), MLR (*p* = 0.294), PLR (*p* = 0.772), NLPR (*p* = 0.115), SII (*p* = 0.798), SIRI (*p* = 0.447), and AISI (*p* = 0.248) (Table 5).

## 4. Discussion

The present study aimed to provide a comparative perspective on the differences between brucella spondylodiscitis and MC1 considering HII. The results disclosed that the HII evaluated herein (NLR, MLR, PLR, NLPR, SII, SIRI, and AISI) were not useful to distinguish between brucella spondylodiscitis and MC1. Therefore, these indices cannot be used to distinguish between the two conditions. However, brucella spondylodiscitis and MC1 were found to be different from each other in pain duration, CRP and ESR levels, involved spinal levels, and platelet count. Thus, these parameters may be used to discriminate between brucella spondylodiscitis and MC1. Nevertheless, this was the first study on the research topic, and its potential limitations should be taken into account.

No study to date has compared brucella spondylodiscitis and MC1 taking HII into consideration. Previous studies have reported that increased HII can be used to predict brucellar arthritis, complicated, and organ-involved brucellosis [35,36,37]. However, these studies were not related to comparison between brucella spondylodiscitis and MC1, and we do not know if they have analyzed baseline (pre-treatment) HII. In the present study, we analyzed baseline laboratory parameters and we found that cases with brucella spondylodiscitis were similar to those without (cases with MC1) in terms of baseline HII. On one hand, brucellosis is associated with pancytopenia [38,39,40], thrombocytopenia [41], and neutropenia [42], but on the other hand, brucella-induced cytopenia is reversible after brucellosis therapy including antibiotics and steroids [41,42,43], and steroids can also lead to leukocytosis and neutrophilia [44,45,46]. Therefore, baseline (pre-treatment) scores of CBC and HII may not be too high and may be different from those at other periods of brucellosis. Indeed, our results demonstrated that patients with brucella spondylodiscitis had a significantly lower platelet count than cases with MC1. Also, although they were not statistically significant, the levels of leukocytes, neutrophils, and monocytes were lower in patients with brucella spondylodiscitis. As a result, due to the hematological cell-reducing effect of brucellosis, baseline CBC-derived HII may not be appropriate for differentiating brucella spondylodiscitis from MC1. Therefore, although HIIs have been suggested as valid indicators of inflammation, infection, and sepsis [47,48,49], this may not apply to brucellosis. On the other hand, we found that patients with brucella spondylodiscitis have lower pain duration, higher levels of CRP and ESR, and a lower platelet count than cases with type 1 MCs. So, these differences can be helpful in the early discrimination of brucella spondylodiscitis from MC1. Nevertheless, to our knowledge, this was the first investigation on the research topic, and potential limitations of the study should be considered.

The potential limitations of the present study should not be overlooked. Although the study’s hypothesis was not supported by the findings, it may provide new insights. For example, it can be suggested that brucellosis may affect HII differently than other infections. Considering the fact that a retrospective design may cause missing data, our study may be within this scope. Some of its features including a single-center trial, a small sample size, partially high female rate, and a limited age range are other potential limitations of the present study that need to be stated. In addition, since there is no research directly related to the subject of the study in the literature, a comparison and discussion of different results could not be made.

## 5. Conclusions

In conclusion, distinguishing between brucella spondylodiscitis and MC1 can present some difficulties. Increased HII can be used to differentiate between infectious and non-infectious conditions, but this may not be valid in brucellosis. Because brucellosis is associated with decreased complete blood cells used in the calculations of HII, this feature of brucellosis and the value of HII to differentiate brucella from other infectious agents can be considered in future studies.

## Figures and Tables

**Table 1 medicina-60-01131-t001:** MRI, histopathological, and histomorphometric characteristics of Modic changes [1,2,3].

	Modic Type 1 Changes	Modic Type 2 Changes	Modic Type 3 Changes
T1 weighted MRI characteristics	Low	High	Low
T2 weighted MRI characteristics	High	High	Low
Histopathological characteristics	Bone marrow edema and inflammation	Fatty degeneration of the bone marrow	Subchondral bone sclerosis
Histomorphometric characteristics	High bone turnover	Low bone formation	Stable sclerosis

**Table 2 medicina-60-01131-t002:** Calculation of hematological inflammatory indices [31,32,33,34].

Index	Calculation
NLR	Neutrophil/Lymphocyte
MLR	Monocyte/Lymphocyte
PLR	Platelet/Lymphocyte
NLPR	Neutrophil/(Lymphocyte * Platelet)
SII	Neutrophil * Platelet/Lymphocyte
SIRI	Neutrophil * Monocyte/Lymphocyte
AISI	Neutrophil * Platelet * Monocyte/Lymphocyte

SII: systemic immune inflammation index; SIRI: systemic inflammatory response index; AISI: aggregate index of systemic inflammation.

**Table 3 medicina-60-01131-t003:** Comparing groups for age, gender, pain duration, CRP, ESR, and spinal levels.

	Brucella Spondylodiscitis (n = 35)	Modic Type Changes (n = 37)	*p*
Age, years	46.17 ± 15.98 (18–64)	44.46 ± 8.78 (28–62)	0.579 *
Gender, F/M	17/18	26/11	0.092 **
Pain duration	57.0 ± 43.51 (10–200) days	8.73 ± 5.98 (1.0–25.0) yrs	<0.001 *
CRP, mg/L	40.86 ± 35.66 (3.0–136.0)	2.65 ± 2.49 (0.17–12.24)	<0.001 *
ESR, mm/h	41.06 ± 21.09 (2–93)	17.68 ± 10.57 (1.0–51.0)	<0.001 *
Spinal levels			0.030 ***
L5–S1	10 (28.57%)	22 (59.46%)	
L4–L5	8 (22.86%)	10 (27.03%)	
L3–L4	5 (14.29%)	1 (2.70%)	
L2–L3	7 (20.0%)	1 (2.70%)	
L1–L2	2 (5.71%)	1 (2.70%)	
Two levels	3 (8.57%)	2 (5.41%)	

CRP: C-reactive protein; ESR: Erythrocyte sedimentation rate; *: The independent *t*-test; **: Fisher’s exact test; ***: The Pearson’s chi-squared test.

**Table 4 medicina-60-01131-t004:** Comparing groups for complete blood cells.

	Brucella Spondylodiscitis (n = 35)	Modic Type 1 Changes (n = 37)	*p*
Leukocyte (10^3^/mL)	6.52 ± 2.13 (3.40–12.80)	7.24 ± 1.85 (3.99–12.25)	0.127 *
Neutrophil (10^3^/mL)	3.70 ± 1.78 (1.20–8.60)	4.05 ± 1.46 (1.72–7.65)	0.366 *
Lymphocyte (10^3^/mL)	3.0 ± 0.66 (1.10–3.50)	2.37 ± 0.68 (1.40–4.43)	0.090 *
Monocyte (10^3^/mL)	0.54 ± 0.26 (0.20–1.60)	0.56 ± 0.24 (0.27–1.50)	0.756 *
Platelet (10^3^/mL)	263.23 ± 72.24(150–423)	297.69 ± 72.26 (177.0–463.0)	0.047 *

The data were presented as mean ± SD (min.–max.), *: The independent *t*-test.

**Table 5 medicina-60-01131-t005:** Comparing groups for hematological inflammatory indices.

	Brucella Spondylodiscitis (n = 35)	Modic Type 1 Changes (n = 37)	*p*
NLR	1.95 ± 1.16 (0.67–5.55)	1.81 ± 0.81 (0.58–5.0)	0.553 *
MLR	0.27 ± 0.11 (0.08–0.55)	0.24 ± 0.09 (0.12–0.44)	0.294 *
PLR	137.77 ± 54.38 (59.09–240.0)	134.27 ± 47.82 (54.28–297.86)	0.772 *
NLPR	0.008 ± 0.005 (0.0–0.02)	0.006 ± 0.003 (0.0–0.01)	0.115 *
SII	527.12 ± 365.04 (112.0–1490.67)	547.49 ± 305.33 (129.18–1795.0)	0.798 *
SIRI	1.12 ± 1.02 (0.21–4.37)	1.03 ± 0.65 (0.34–3.20)	0.447 **
AISI	302.87 ± 290.95 (44.80–1192.53)	317.36 ± 239.01 (66.05–1148.80)	0.248 **

NLR: neutrophil/lymphocyte; MLR: monocyte/lymphocyte; PLR: platelet/lymphocyte; NLPR: neutrophil/(lymphocyte * platelet); SII (neutrophil * platelet/lymphocyte): systemic immune-inflammation index; SIRI (neutrophil * monocyte/lymphocyte): systemic inflammatory response index; AISI (neutrophil * platelet * monocyte/lymphocyte): aggregate index of systemic inflammation. *: The independent *t*-test, **: The Mann–Whitney *U* test.

## Data Availability

The data sets used in the present study are available from the corresponding author upon reasonable request.

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
