# Peer review of "Can Hematological Inflammatory Indices Be Used to Differentiate Modic Type 1 Changes from Brucella Spondylodiscitis?"

_medicina, 2024, doi:10.3390/medicina60071131_

Round 1

Reviewer 1 Report

Comments and Suggestions for Authors

Overall, the article is well conceived and could be of interest to the readership of this journal. 

I have following comments which need to be carefully addressed before considering this article for publication.

1. The authors have described that serological, microbiological and radiological based diagnosis was used to confirm the disease. Please elaborate the exact test used to diagnose the disease.

2. The authors have described that there were no significant difference in demographic characteristics of the selected patients. however, there seem the tendency of proportionate higher recruitment of females (p value around 0.09). This could be addressed by using proportionate sampling and higher sample size. Furthermore, how could this affect the results need to be addressed in discussion.

3. For recruitment of the population, the inclusion and exclusion criteria may be presented to better facilitate the readership of this journal.

4. What type of medication was being used in selected groups and how could this impact haematological parameters.

5. The results need to be improved in terms of presentation.

6. To increase the robustness of results, the sample size may be improved.  

Author Response

Thank you for your positive evaluations.

We have carefully addressed your comments and incorporated into revised article.

1.The sentence "A positive culture from any specimen was considered a definitive diagnosis of brucellosis." has been added in the Methods section.

2. You may be right in your suggestion, but we cannot fulfill this suggestion, especially since brucella spondylodiscitis cases are rare and there is a risk of losing time in academic processes. However, it should be taken into consideration that this is the first study on the topic and the possible effect of partially high female rate on the resuls has been stated in the limitations.  

3. The inclusion and exclusion criteria have been presented to better facilitate the readership of this journal.

 4. In the present study, we analyzed baseline (pre-treatment) laboratory values. Therefore, it is not possible fort he medication used to affect the haematological parameters analyzed.

5. The results have been redesigned and improved in terms of presentation.

6. You may be right in your suggestion, but we cannot fulfill this suggestion, especially since brucella spondylodiscitis cases are rare and there is a risk of losing time in academic processes. However, it should be taken into consideration that this is the first study on the topic and the small sample size has been stated in the limitations.

Reviewer 2 Report

Comments and Suggestions for Authors

The manuscript entitled "Can hematological inflammatory indices be used to differentiate Modic type 1 changes from Brucella spondylodiscitis?" promises to provide new diagnostic aid for differentiating Modic type 1 changes from Brucella spondylodiscitis. Still, it failed to identify a clear test or aid. The major problem is the allocation of patients with brucellosis. How has brucellosis been confirmed in these patients? Have isolation and identification, the gold standard test been performed, or only serological testing? There can be changes in HII in other infectious causes also. So, how to rule out these changes? Since this is a retrospective study a larger dataset with other infectious diseases will help to identify a marker for differentiation. Presently the data presented is very naive. 

Minor changes

Introduction:

MC13- should be MC3

autoimmune reaxion- Check the sentence for its correctness

Comments on the Quality of English Language

The quality of English is readable

Author Response

Thank you for your evaluations and suggestions.

Diagnoses of brucella spondylodiscitis and MC1 were supported by microbiological, serological, and radiological diagnotic tools. A positive culture from any specimen was considered a definitive diagnosis of brucellosis. The sentence "A positive culture from any specimen was considered a definitive diagnosis of brucellosis." has been added in the Methods section. Also, in the Methods section, it has been stated that "…other infectious spondylodiscitis types than brucella, other types of Modic changes than MC1, and other non-infectious conditions such as rheumatic spondylodiscitis (ankylosing spondylitis or Andersson lesion) were excluded from the study."

Minor changes that you have mentioned have been corrected. Thank you for your attention.

MC13 was corrected as MC3

autoimmune reaxion was corrected as autoimmune reaction

Thank you for your positive evaluation.

Reviewer 3 Report

Comments and Suggestions for Authors

The manuscript by Sah and Baran entitled "Can hematological inflammatory indices be used to differentiate Modic type 1 changes from Brucella spondylodiscitis?" compares inflammatory parameters between patients with Brucella spondylodiscitis and Modic type I. The manuscript is interesting because a differential diagnosis is necessary, given the clinical similarity between the two conditions in endemic areas of brucellosis. That is the reason why the information presented in the manuscript is practical. However, the authors must make several changes to the manuscript. Since the introduction, the authors should clarify that the study is focused on detecting the differences in the inflammatory parameters between the clinical conditions and clearly define this objective. In the discussion, the authors should rewrite the section where they discuss articles 35, 36, and 37; already, these articles compare inflammatory conditions of patients with brucellosis vs. healthy volunteers or patients with complications vs. patients with no complications. After all, the authors do not find differences between the analyzed populations because both present significant inflammatory alterations. That is why I recommend rethinking a new title and focusing on highlighting the differences between brucella spondylodiscitis and Modic type I, as evidenced in their work.   

Author Response

Thank you for your positive evaluations and helpful suggestions.

You are right. In the introduction, the following sentence has been added to the last paragraph of Introduction section: "Therefore, the present study is focused on detecting the differences espesially in the simple clinical and inflammatory parameters between brucella spondylodiscitis and MC1."

You are very right. In the discussion, as you suggested, we have rewrite the section discussing articles 35, 36, and 37. Since no study to date has compared brucella spondylodiscitis and MC1 taking HII into consideration, we discussed indirectly related articles. The section has been corrected as more understantable. Please see red colored parts and all discussion.

You may be right in your suggestion. However, considering specific features and popularity of hematological inflammatory indices (HII) and brucellosis-related hematological changes, we focused HII roles in differentiate Modic type 1 changes from Brucella spondylodiscitis. Therefore, we used the current title.

Round 2

Reviewer 1 Report

Comments and Suggestions for Authors

My queries have been addressed and I am satisfied with the updated version.

Reviewer 2 Report

Comments and Suggestions for Authors

The authors have addressed the comments

Reviewer 3 Report

Comments and Suggestions for Authors

Accept in present form